# Surveillance of *Legionella* spp. in Open Fountains: Does It Pose a Risk?

**DOI:** 10.3390/microorganisms10122458

**Published:** 2022-12-13

**Authors:** Ioanna P. Chatziprodromidou, Ilektra Savoglidou, Venia Stavrou, George Vantarakis, Apostolos Vantarakis

**Affiliations:** Environmental Microbiology, Department of Public Health, Medical School, University of Patras, 265 04 Patras, Greece

**Keywords:** *Legionella*, water, open fountain, risk, surveillance, public health

## Abstract

Clusters of outbreaks or cases of legionellosis have been linked to fountains. The function of fountains, along with their inadequate design and poor sanitation, in combination with the warm Mediterranean climate, can favor the proliferation of *Legionella* in water systems. Public fountains in Mediterranean cities may pose a significant risk for public health due to the aerosolization of water. Nevertheless, few studies have been conducted on *Legionella* and the risk of infection in humans through fountains. In our study, the presence and quantity of *Legionella* spp. in fifteen external public fountains were investigated. Two samplings were performed in two different periods (dry and wet). Sixty samples were collected, quantified and analyzed with a culture ISO method. The operation of all fountains was evaluated twice using a standardized checklist. In accordance with their operation, a ranking factor (R factor) was suggested. Finally, based on these results, a quantitative microbial risk assessment was performed. Thirty water samples taken from the fountains (100%) during the dry sampling period were positive for Legionella (mean log concentration: 3.64 ± 0.45 cfu/L), whereas 24 water samples taken from the fountains during the wet period were *Legionella*-positive (mean log concentration: 2.36 ± 1.23 cfu/L). All fountains were classified as unsatisfactory according to the checklist for the evaluation of their function. A statistically significant correlation was found between *Legionella* concentration and the assessment score. The risk of *Legionella* infection was estimated in both periods, with higher risk in the dry period. The surveillance and risk assessment of *Legionella* spp. in the fountains of Patras confirmed a high prevalence and a high risk to public health.

## 1. Introduction

*Legionella* is a genus of Gram-negative bacteria that causes legionellosis in humans. Legionellosis has two distinct clinical forms: Legionnaires’ disease and Pontiac fever [1]. The most severe form of Legionnaires’ disease (LD) is pneumonia, a severe multisystem disease, while Pontiac fever is an influenza—similar to a cold [2,3,4,5]. The *Legionella* genus consists of 59 species and at least 72 serogroups, about half of which have been clinically observed to be pathogenic for humans. Nonetheless, the majority of *Legionella* spp. are considered to be facultative pathogens [4,5,6,7]. *Legionella pneumophila*, the causative agent of the outbreak in Philadelphia in 1976, is the etiological agent of about 90% of LD cases and one of the most studied species [5,6]. The fatality rate of LD was found to be about 10% [8]. Mortality rates are highly variable, ranging from 1% to as high as 80% [5]. The mortality rates can be largely attributed to the following determining factors: the patient’s treatment, the promptness of the treatment and whether the disease is considered to be sporadic, nosocomial or part of a large outbreak. On the other hand, information concerning Pontiac fever is limited, probably due to the similarity of its symptoms with the common flu [5]. 

*Legionella* bacteria are ubiquitous in natural water environments and have a worldwide distribution. They are part of the natural microbial flora of many natural aquatic environments where their concentrations are usually low, representing a minor component (<1%) of the residential bacterial population [7,9,10]. They can be found in streams, rivers, ponds, lakes, mud and moist soil, where they can survive for a long time [5]. They can withstand temperatures of 0–68 °C [5] and reproduce at 25–42 °C [7]. *Legionella* bacteria enter manmade water systems from freshwater environments. With an optimal growth temperature of 35 °C, they can proliferate in thermal water supply systems, such as whirlpool spas, hot water systems, air-conditioning cooling towers, taps and shower heads, and reach high concentrations [5,9]. 

In addition to the temperature, several factors enable *Legionella* to multiply, or at least persist, in manmade water systems under a wide range of environmental conditions. These factors are: growth in and protection by free-living amoeba, biofilm formation, growth at low oxygen concentrations and in low-nutrient environments, the ability to enter the viable but nonculturable state and disinfectant tolerance when hosted by biofilm or amoeba [11,12,13]. By properly maintaining a system and applying standard sanitizing measures, biofilm formation can be avoided, and nutrients and amoebas can be kept at lower levels. 

The environment is the only source of infection for legionellosis since this infection is not documented to be transmitted from person to person. The transmission of *Legionella* from a contaminated source, natural or manmade, may occur in two different ways: (a) through inhalation of aerosols or (b) through aspiration of fluids [2]. The most common artificial water sources for infection—and also the most common confirmed causes of outbreaks—are cooling towers, baths, wastewater, room humidifiers, air conditioning systems and fountains [14]. Pontiac fever and outbreaks of Legionnaires’ disease in hospitals [15,16], squares [17], hotels [18,19,20] and restaurants [21,22] have been linked to decorative fountains found in the premises of these locations. In most cases, inadequate maintenance or errors in the maintenance procedure are the causes of increases in *Legionella* concentrations in the water of fountains.

In this research, we conducted an environmental study of 15 public open fountains in Patras city, Greece, which is the third largest city in Greece and the regional capital of Western Greece. There were no previous epidemiological data or cases correlated with fountain exposure. The aims of the study were: (a) to investigate the frequency and magnitude of *Legionella* colonization in fountains, (b) to evaluate and classify the fountains’ operation, (c) to correlate *Legionella* concentrations with the operation of the fountains and (d) to assess the risk of *Legionella* infection. To the best of our knowledge, this is the first risk assessment study of *Legionella* infection based on water analyses conducted on fountains.

## 2. Materials and Methods

### 2.1. Evaluation of Fountains

In July 2018, the first communication with the relevant local authorities responsible for public fountains was initiated. A laboratory technician visited all the fountains accompanied by the municipality manager. The locations of the open public fountains of the city and their design, timetable of operation and maintenance procedures were recorded. More critical issues, such as the origin of the water, the disinfection of the water, the frequency of cleaning and the possible use of filters, were also discussed. The discussion revealed that the water in the fountains comes directly from the municipal water distribution system and no additional disinfectants or filters are used. 

On 27 July 2018 (dry period) and 6 November 2018 (wet period), “on-the-spot evaluation checks” of the fountains were carried out. For this purpose, it was considered appropriate to use the questionnaire recommended by the Greek National School of Public Health (NPHS) (Appendix A). The checklist was developed by the Environmental Health Surveillance to assess several targets of public health importance, including LD concentrations during the Athens 2004 pre-Olympic and Olympic period [23]. Its system control points (SCPs) are based on the requirements of the International Standardization Organization (ISO), National and European legislation and the World Health Organization guidelines [17]. At first, basic elements, such as the code of the fountain, the check-in date, the check-in time, the date of the last maintenance or inspection, and the name of the person who work on or inspected the fountains, were completed in the questionnaire. Afterwards, the 17 SCPs included in the checklist were evaluated and recorded. SCPs 2–9 were evaluated through visual observation; 1, 10, 11, 15 and 16 were evaluated after communication with the relevant local authorities; and 11–14 and 17 were measured in a laboratory (Appendix A). SCPs 15 and 16 referred to chemical and microbiological analyses conducted by the relevant local authorities during the last year. SCP 17 referred to the samples collected for microbiological analysis by the laboratory during the “on-the-spot evaluation check” of the fountains. The system control points were, in short, the operation and maintenance of the system; the absence of obvious faults, leaks, stagnation and regression; the absence of corrosion, salts, microbiological growth and algae; cleaning and disinfection; pH; temperature; and the condition of the filters.

During the evaluation checks, each SCP was used as a score item as it was designed by the NPHS. A positive answer for an SCP added no points and a negative answer subtracted points according to the severity of the violation. The score values varied from −1 to −3. The total score was calculated for each fountain for both dry and wet periods. Afterwards, each fountain was classified in both periods according to its score in one of the following three categories: satisfactory system operation (0 to −2 points), partially satisfactory system operation (−3 to −5 points) and unsatisfactory system (lower than −5 points). SCP 17 was not included in the score for the fountains. 

### 2.2. Sample Collection

Double sample collection was performed during each period (dry and wet periods). During the dry period, samples were collected on 27 July 2018 and 1 August 2018, and during the wet period, on 6 November 2018 and 9 November 2018 (60 samples in total). Since the water supply company in Patras carries out daily chemical and microbiological quality controls, we considered that sample collection from the central network was not necessary in this study. One liter of water was collected from the center of each fountain (60 cm away from the edge) in sterile glass bottles according to CDC guidelines (routine testing of Legionella). The bottles contained sodium thiosulfate to neutralize any residual chlorine. The samples were transferred to the laboratory in refrigerators containing ice packs. A microbiological analysis was performed within 8 h, as recommended by Greek legislation 3282, 2017. The samples were kept at 5 ± 3 °C until the analysis was performed.

### 2.3. Physicochemical Analysis

The water temperature, the air temperature and the air humidity were measured at the time of the sample collection. The sanitary situation of the fountains was recorded (clean or not). Since the fountains use water from the central network without further disinfection, residual chlorine was the only disinfectant measured. The chorine and pH measurements were conducted during the sampling. 

### 2.4. Microbiological Analysis Method

A microbiological analysis for the measurement of *Legionella* spp. was performed at the Public Health Laboratory of the Medical Department of the University of Patras in accordance with ISO 11731:2017. In brief, one liter of water sample was filtered through a 0.2 μm polyethersulfone filter (Pall Corporation). The membrane filter was cut into pieces, transferred into 10 mL phosphate buffered saline (Sigma) and mixed vigorously by vortex for 2 min to wash the microorganisms from the membrane. The sample was divided into three portions, which were treated with either heat (50 °C for 30 min in a water bath) or an acid solution of HCl and KCl (1 volume sample + 9 volumes acid, mixed and left to stand for 5.0 ± 0.5 min) or left untreated, as suggested by ISO. From each portion (untreated, heat-treated, acid-treated), 0.1 mL was spread in BCYE agar (Oxoid) and GVPC agar (Oxoid). The plates were incubated at 36 ± 2 °C for 7–10 days in a humid atmosphere. The plates were inspected every day to identify overgrowth. Suspected colonies were counted at the end of the incubation period. Subculturing was carried out from the plates that seemed to have the highest counts of presumed colonies of *Legionella* spp. per water volume. Three colonies of each type were subcultured in BCYE agar and BCYE without cysteine agar (Oxoid). Confirmed colonies were considered as those colonies that grew on BCYE but did not grow on BCYE without cysteine. The number of colony-forming units of *Legionella* spp. in the initial water samples was calculated based on the plate with the maximum number of confirmed colonies per water volume, in accordance with ISO 8199. The limit of detection (LOD) for the method was 100 cfu/L and samples with fewer colony-forming units per liter were negative for *Legionella* spp.

### 2.5. Quantitative Microbial Risk Assessment (QMRA)

Human exposure to ornamental fountains may lead to health risks when the water is contaminated with pathogens. The potential health risks associated with exposure to *Legionella* from fountains were calculated using the quantitative microbial risk assessment (QMRA) method. The QMRA model consists of the following four steps: (1) hazard identification, (2) exposure assessment, (3) dose–response modeling and (4) risk characterization [10,24]. 

#### 2.5.1. Hazard Identification

For the conduction of the QMRA, all *Legionella* spp. (i.e., not only *Legionella pneumophila*) were considered as hazardous. The ESCMID Study Group for *Legionella* Infections (ESGLI), with regard to conducting a risk assessment, clearly states that: “While *Legionella pneumophila* serogroup 1 is the strain most associated with causing community-acquired cases, the detection of other strains of *Legionella* in routine testing does not necessarily mean the risk is reduced. This is because during routine testing only a minority (maybe as few as one or two colonies on a culture plate) will be tested”. In parallel, in the action levels that are proposed following *Legionella* sampling, the concentrations of the entire genus are evaluated [25].

All the public fountains studied were located in the city of Patras and, specifically, in public places, such as squares. Fountains were found in the vicinity of some squares and playgrounds, which is where Greek families spend their leisure time, especially during spring and summer [26]. Most squares have coffee shops with tables and chairs all around them. All squares have benches, which function as meeting points, especially for teenagers and students. The concentration of *Legionella* in the air (C_air_) from each fountain was estimated using the established emission factor (EF) specified for this exposure scenario, and the average *Legionella* spp. concentrations in water (C_water_) were taken into account in the analysis of the sampled points. The emission factor (EF) is the ratio between the concentration of bacteria in air and their concentration in water. This factor was used to estimate the bacterial concentration in air at a known water concentration. The emission factor (EF) for fountains was calculated following de Man et al. [10,27].

#### 2.5.2. Exposure Assessment and Inhalation Rate

The factors that determine the inhalation exposure dose (IED) are: the concentration of *Legionella* in the air (cfu/m^3^) (C_air_), the duration of the exposure (ED), the inhalation rate (IR) and the respirable aerosol retention rate (RR) (Table 1) [10,28].

The duration of the exposure was considered to be 330 min weekly based on the study by Sales Ortells et al. (2014) [29]. For a large percentage of the population, this duration of exposure is a realistic estimation of the time spent around fountains, since they are placed in the squares of the city.

**Table 1 microorganisms-10-02458-t001:** QMRA parameters (mean values).

Parameter		Value	Unit	Reference
EF ^a^	Emission factor for fountains	8.6 × 10^−9^	L/m^3^	de Man et al. (2014) [27]
IR ^b^	Inhalation rate	1.05	m^3^/h	Schoen et al. (2011) [30]
RR	Retention rate	0.5		Armstrong and Has (2007) [31]
ED	Exposure duration	330	min/week	Sales-Ortel et al. (2014) [29]

a: EF from similar water operation system; b: the average inhalation rate of middle—aged man and women.

To calculate the inhalation exposure dose, it was essential to estimate the *Legionella* concentration in the air. For this purpose, we used the emission factor (EF), which is the ratio between the bacterial concentration in the air (C_air_) and the bacterial concentration in the water (C_water_) (Table 1) [10,28,30]. No data were available for the emission factor for fountains in the literature. This factor was used to estimate the bacterial concentration in air at a known water concentration. This was the justification for why we used the emission factor value for a water system with a similar operation as fountains, as used by de Man et al. (2014) [10,27].

The inhalation exposure dose was estimated based on the *Legionella* concentrations in the air, the duration of exposure, the inhalation rate, and the respirable aerosol fractional retention rate (Table 2. Equation (2)) (Armstrong and Haas, 2007a). The model parameters used and corresponding references are presented in Table 1. Due to the lack of data, we could not quantitatively assess the effects of the distance from fountains that were not encountered in our analysis.

#### 2.5.3. Dose–Response Modeling

The risk of *Legionella* infection was assessed separately for each of the 15 fountains and the average concentrations were estimated for each period. The risk of infection was assessed based upon the exponential model (Table 2, Equation (3)) [28]. Average concentrations were calculated for each of the time periods using the dose–response relationship for Legionnaires’ disease and the estimated inhalation exposure. Concerning the disease transmission model, infection was considered more protective and beneficial as a parameter than other endpoints, such as disease and death (Buse et al., 2012). Rw(d) indicates the predicted risk at a given weekly dose d (cfu) multiplied by the *Legionella* infection-risk model parameter for γ = 0.06 (1/cfu) (Armstrong and Haas, 2007b). In addition, the 95% confidence intervals for the γ parameter (γ = 0.039 and γ = 0.131) were used to calculate the minimum and maximum risks of infection. Confidence intervals for this parameter were taken from the QMRA wiki, maintained by the Center for Advanced Microbial Risk Assessment at Michigan State University (http://qmrawiki.canr.msu.edu/index.php/Legionella_pneumophila:Dose_Response_Models; accessed on 29 October 2022).

### 2.6. Risk Characterization

In the evaluation of the risk of microorganisms in the water or the air, each day represented an exposure event; therefore, the input dose in the dose–response model was the average number of microorganisms per week. Duration data were obtained from the literature; specifically, weekly exposure duration was based on the durations reported by Sales-Ortells (2014). Each sampling date in our study was assumed to constitute an exposure event. The cumulative annual risk (in 2018) was calculated using Equation (4) from Table 2. R_a_(d) indicates the periodic cumulative risk for a given R_w_(d), and n is the total number of exposure events (26 weeks for each period).

### 2.7. Risk Assessment 

#### 2.7.1. Bacterial Water in Air Coefficient Approach 

*Legionella* spp. water concentrations (C_water_) were assessed using the relative points for fountains (Figure 1, Figure 2 and Figure 3). Using an approximation of the water–air bacterial partition coefficient, these values were employed to calculate the bacterial concentrations in air (C_air_), as described earlier in this section. The equation used in this study was C_air_ = C_water_ × 8.6 × 10^−9^.

#### 2.7.2. Inhalation Exposure Dose Estimation

Weekly inhalation exposure doses (cfu) for the fountains were obtained based on the *Legionella* concentrations in the air, the exposure time and the inhalation rate and retention rate of respirable aerosols, as described earlier in this section. The calculated inhalation exposure doses (IEDs) for each fountain were based on the following equation: IED = C_air_ × 330 × 0.5 × 1.05. They are presented in Table 4.

#### 2.7.3. Statistical Analysis

A descriptive data analysis was carried out using Microsoft Excel and statistical analyses were conducted in R version 1.2.5042. The Mann–Whitney test was applied to compare *Legionella* spp. concentrations between dry and wet seasons. Spearman correlation coefficients and *p*-values were calculated for each input parameter and the probability of *Legionella* spp. disease; i.e., the model output. The results were statistically significant when the *p*-value was <0.05 and highly significant when the *p*-value was <0.001.

## 3. Results

### 3.1. The Concentrations of Legionella

The sampling points and *Legionella* concentrations are shown in Figure 1. All water samples taken from the fountains during the dry sampling period (100%) were positive for *Legionella*, with a mean log concentration of 3.64 ± 0.45 cfu/L (range: 3.00–4.55 cfu/L). Eighty percent of the water samples taken from the fountains during the wet period were *Legionella* positive. The mean log concentration was 2.36 ± 1.23 cfu/L (range: 0.00–3.80 cfu/L) (Figure 2). 

The concentration of *Legionella* spp. varied according to the period (Mann–Whitney test, *p* < 0.001) (Figure 3). A statistically significant positive correlation between the concentration of *Legionella* and water temperature (Figure 4), humidity and air temperature was found (Spearman test, *p* < 0.001 r = 0.728, *p* < 0.001 r = 0.589, *p* < 0.001 r = 0.611) (Appendix B, Figure A1 and Figure A2). On the other hand, no statistically significant correlation was found between the concentration of *Legionella* and pH values (Spearman test *p* = 0.372, r = −0.117) (Appendix B, Figure A3). The *Legionella* concentration findings did not depend on the fountain cleanliness (Mann–Whitney test *p* = 0.096).

### 3.2. Physicochemical Results

The fountains operate through the water network. Nevertheless, residual chlorine was not present in any of the fountains during either period. The mean pH values were 8.18 ± 0.32 (min. 7.55, max. 8.59) during the dry period and 8.22 ± 0.22 (min. 7.50, max. 8.57) during the wet period. No filters were used in the fountains. No chemicals were used for the salts in the pipelines and pumps. The mean water temperatures were 26.27 ± 2.25 °C (min. 21.1 °C, max. 30.9 °C) during the dry period and 15.80 ± 1.21 °C (min. 13.02 °C, max.18.30 °C) during the wet period. The mean humidity values were 63.83 ± 3.94 (min. 52, max. 69) during the dry period and 55.07 ± 8.25 (min. 41, max. 73) during the wet period. The mean air temperatures were 32.03 ± 2.73 °C (min. 26.0 °C, max. 36.0 °C) during the dry period and 21.28 ± 3.15 °C (min. 13.02 °C, max. 18.30 °C) during the wet period. In many cases, the development of salts, microbiological growth and the presence of algae were obvious. No chemical tests or microbiological examinations had been performed in the previous year by the local water authorities (points 15 and 16 in Table 3).

### 3.3. Classification of the Operation of Fountains

The compliance with the questions from the checklist is shown in Table 3. All fountains had scores lower than −6 and their operation was classified as unsatisfactory (Figure 5). A statistically significant negative correlation was found between the *Legionella* concentration and the assessment score (Spearman’s test, *p* < 0.031, r = −0.395).

### 3.4. Risk Characterization

The period risk levels are reported in Table 4. The seasonal and annual risk characterizations for both dry- and wet-period sampling are summarized in Figure 6.

## 4. Discussion

In the present study, the microbiological analysis of the water in the open fountains in Patras confirmed a high prevalence of *Legionella* species. These city fountains were all colonized by *Legionella* in the dry period, while 80% of them were colonized in the wet period. The concentrations of *Legionella* varied by season, with higher concentrations during the dry period, as expected. Warm and humid weather helps *Legionella* to survive and proliferate [11]. Higher concentrations of the pathogen during the dry period compared to the wet period can easily be explained by the differences in the water temperature between the two periods, since a statistically significant positive correlation between *Legionella* concentrations and water temperatures was found in our study. The water temperature of the fountains was within the optimal growth temperature range for *Legionella* [5,6,9] during the dry period, which enabled the bacteria to proliferate. Lower water temperatures during the wet period resulted in lower concentrations of *Legionella* in the water of the fountains. Moreover, the estimated risk values varied considerably between seasons, and the risk levels increased by up to two orders of magnitude in summertime compared to the winter months. It should be taken into account that *Legionella* spp. concentrations in both periods may have been higher than those measured due to the ability of *Legionella* to enter the viable but nonculturable state when stressful environmental conditions exist.

No statistically significant correlation between *Legionella* concentrations in the fountains and pH was found (r = 0.117), which was in agreement with the study by Fragou et al. (2012), in which there was no significant correlation between the *Legionella* concentration in the water distribution systems of hotels and hospitals and pH [7], and the study by Papadakis et al. (2018), in which no significant correlation between *Legionella* colonization and pH values was found in samples from the recreational and garden areas of hotels [4]. However, the study by Kyritsi et al. (2018) found that pH ≥ 7.45 increased the risk of colonization of cold-water supply systems by *Legionella* spp. [32]. “pH values between 7.0 and 8.0” was the item in the questionnaire used to assess the stability of chlorine in the water. However, since the water in the fountains was without residual chlorine, this limit was meaningless. Nevertheless, the pH values of Patras’ fountains (mean/L 8.2 ± 0.27) may have been one of the factors that enhanced the colonization of the systems.

The operation of all fountains was evaluated as unsatisfactory. A statistically significant negative correlation between *Legionella* concentrations and the evaluation score was found (r = −0.395). Hadjichristodoulou et al. (2004) found no significant association between unsatisfactory inspection results for fountains and colonization by *Legionella*. This could be explained by the fact that few of the fountains in that study were colonized (3/134) and 76% of the fountains were evaluated as unsatisfactory [23] versus 100% in our study. That indicates that Patras’ fountains lack proper maintenance, do not operate satisfactorily and, at the same time, are colonized in high percentages by *Legionella*, three facts that appear to be related to the fact that unsatisfactory operation causes colonization by this bacterium. In the study by Papadakis et al. (2018), 3 of the 45 fountains were found to be colonized with *Legionella*, as was also found in our study [4].

The repeated-measures Mann–Whitney test showed that the annual risk varied significantly between the dry and wet seasons. The highest rates of risk were associated with fountains during the dry period. Our results showed that the prevalence of *Legionella* in the water system and the resulting risk of infection were time-dependent and could vary significantly, even for neighboring fountains. These findings should be taken into account with regard to public health regulations. We recommend consideration of the different seasons when creating *Legionella* monitoring plans for fountains in Patras, Greece or other countries with a Mediterranean climate. Therefore, routine sampling and control measures should be systematically carried out during the dry season, when the infection and illness risks are significantly higher. A range of Legionella concentrations have been observed in water distribution systems, and the risk predicted by QMRA may be incorrect if a large proportion of the current *Legionella* species are dissimilar in their virulence characteristics [33].

The main limitations of this QMRA framework, as with any overview, included (a) methodological problems, which led to very limited data being derived from the urban city through a limited sampling procedure (only twice per period); (b) the input assumptions for the QMRA model; and (c) the fact that no probability functions were used and, for this reason, no variability or uncertainty were accounted for in the model applied. Moreover, as deterministic estimates are always higher than stochastic ones, this may have led to unnecessary and expensive treatment procedures detrimental to environmental, financial and social gains.

Local authorities could make use of expertise and guidelines from other countries. In Singapore, legislation for the control of *Legionella* bacteria in cooling towers and water fountains was enacted in 2001. Concerning fountains, the owner is obligated to keep proper records and ensure that the fountain is in a state of repair and free from algae, scale, dirt, sludge, slime and foreign matter. Inspections for cleanliness, physical defects, organic fouling and physical debris are conducted once a week. In addition, flushing, cleaning and disinfection are carried out once a month [3].

Legionnaires’ disease is a serious and potentially life-threatening disease. The population potentially exposed to this type of water system comprises about 8000 people, and some of them are elderly people, smokers and immunocompromised individuals, who are particularly susceptible [5,34]. Therefore, these results should be taken into account and preventive measures should be implemented, such as disinfecting the water of the distribution system [10].

During the evaluation checks of the fountains in both periods, no residual chlorine was found in the water samples, and none of the fountains had a filter. We recommend the installation of an independent disinfectant device in each public fountain in Patras [15,16]. Since the residual chlorine contained in water mains evaporates immediately and it is impossible to maintain values between 0.4 and 0.7 mg/L, an independent disinfection device (for example, an ultraviolet disinfection device) could be the solution to the disinfection problem in fountains. The use of filters could promote additional growth and sporadic accidental release of large doses, provided that its maintenance protocol (washing, removal of particles, disinfection and replacement) is appropriate.

Both measures applied in combination could reduce *Legionella* spp. concentrations in the water of fountains to acceptable limits.

In this study, the health risks stemming from exposure to *Legionella*-contaminated aerosols were assessed using the QMRA approach. The findings highlighted the need for proper and regular treatment and disinfection of drinking water systems to maintain water safety and quality. Moreover, the calculated risk values varied significantly between seasons, and the risk levels increased by up to two orders of magnitude in the summer compared to the winter. Thus, in countries with a Mediterranean climate, we recommend taking the seasons into account when designing *Legionella* monitoring plans and regulating drinking water networks. Concentrating sampling and monitoring efforts in the summer months will allow for more accurate assessment of the risks posed by the presence of *Legionella* in drinking water systems. To address the high prevalence of Legionella in fountains, several actions can be proposed for the health inspectorates of municipalities: routine cleaning; following manufacturer recommendations; monitoring critical water parameters, such as temperature and residual disinfectants, at least weekly; automating disinfectant feeding and monitoring systems, if possible; applying algaecide as needed; avoiding prolonged idle periods; and running decorative fountains at least daily. The frequency of these activities depends on the environmental conditions present in the area where the decorative fountain is located and its design. Furthermore, legislative initiatives should be implemented by national/EU authorities to introduce the monitoring of Legionella into routine monitoring of water in fountains.

## 5. Conclusions

To the best of our knowledge, the present study is the first to evaluate open fountains for *Legionella* risk and correlate *Legionella* concentrations with a score evaluation for the operation of fountains.

All fountains were colonized by *Legionella* in the dry period and 80% were colonized in the wet period. The Mediterranean climate, the fountains’ unsatisfactory operation, the lack of residual chlorine in the water and the absence of filters are probably the causes of the high prevalence of *Legionella* in the fountains of the city.

Although there is high risk of *Legionella* infection from the fountains in Patras, no cases have been reported. This fact can be explained by either the underestimation of the disease due to a lack of surveillance or the possible incidence of cases of Pontiac fever.

To reduce the risk for public health, the bacterial concentrations should be maintained within low limits. An independent disinfectant device combined with the use of a filter would reduce *Legionella* concentrations directly and indirectly, and inspections to evaluate the fountains’ operation and cleanliness, as well as the disinfection of the water, should be performed at regular intervals along with microbiological analyses.

## Figures and Tables

**Figure 1 microorganisms-10-02458-f001:**
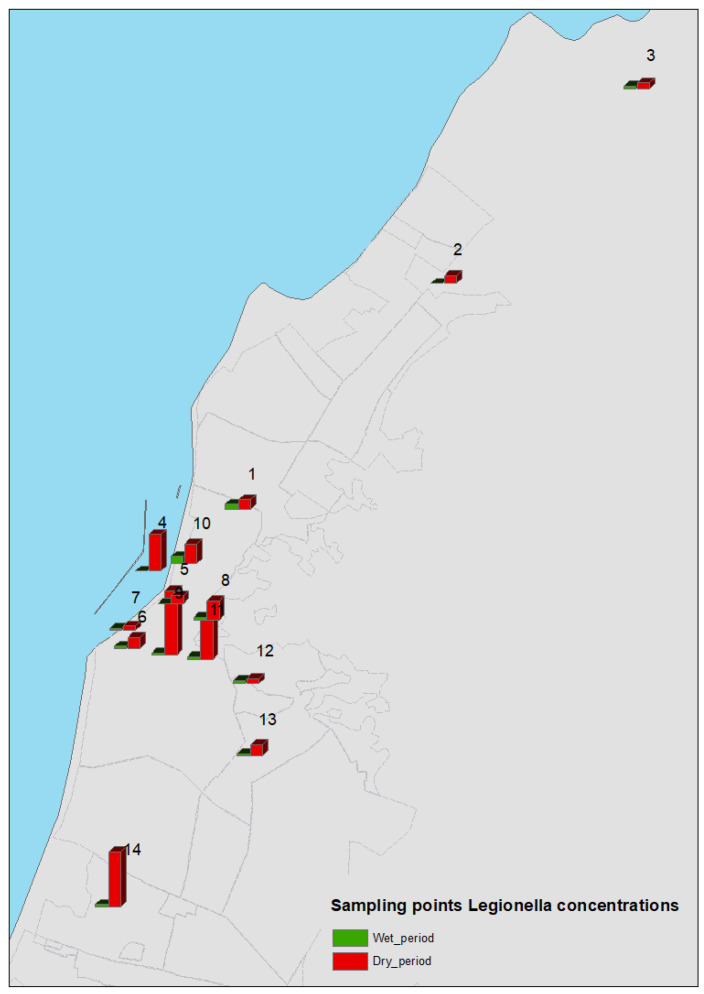
GIS map with sampling points and *Legionella* concentrations. The red columns correspond to the *Legionella* spp. concentrations in the dry period and the green columns to the *Legionella* spp. concentrations in the wet period.

**Figure 2 microorganisms-10-02458-f002:**
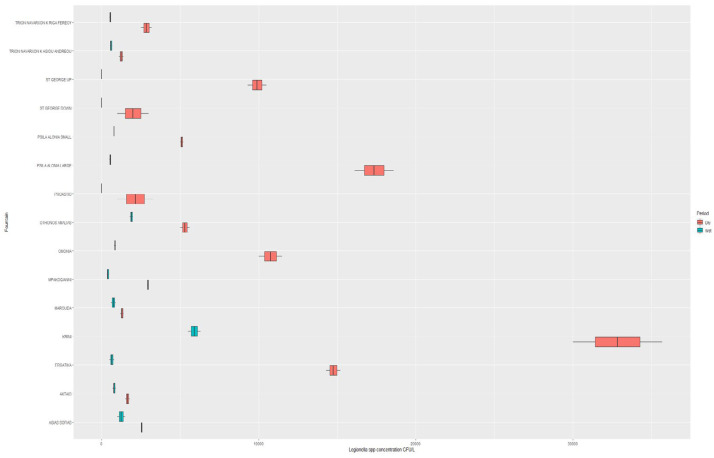
*Legionella* spp. concentrations (on a logarithmic scale) for each fountain per period.

**Figure 3 microorganisms-10-02458-f003:**
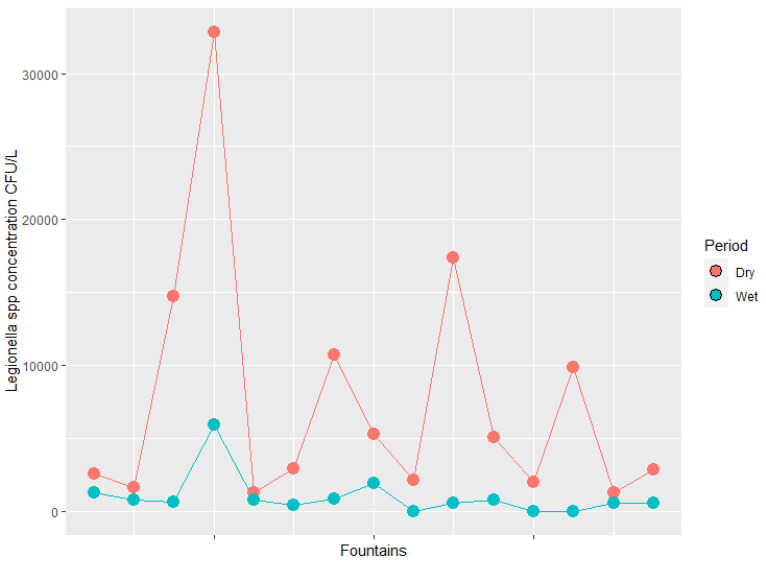
*Legionella* spp. concentration (on a logarithmic scale) correlated with season.

**Figure 4 microorganisms-10-02458-f004:**
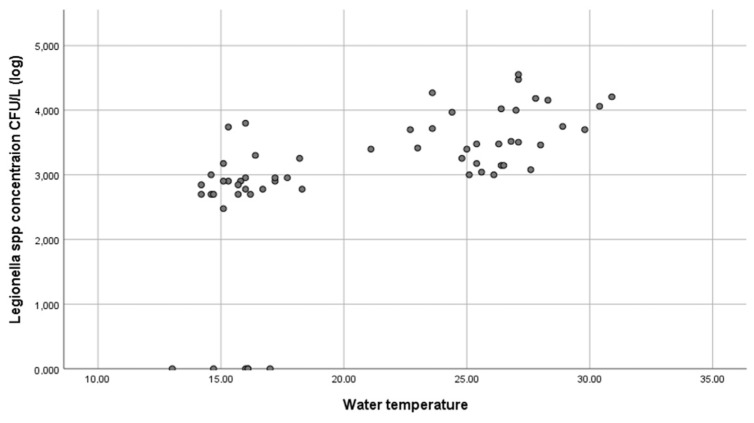
*Legionella* spp. concentration (on a logarithmic scale) correlated with water temperature.

**Figure 5 microorganisms-10-02458-f005:**
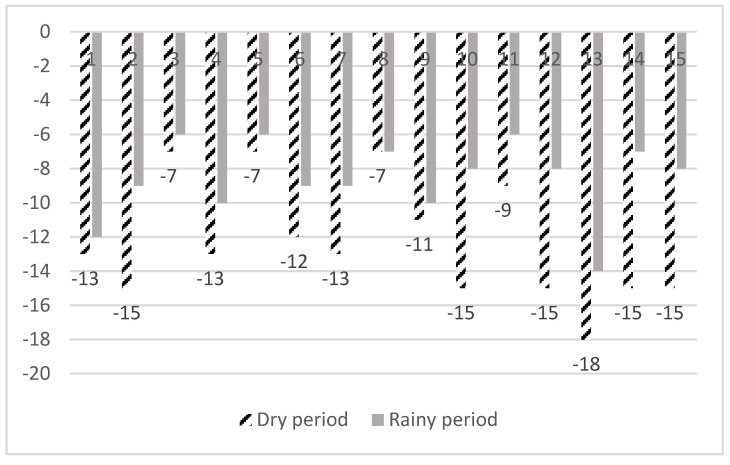
Scores of evaluations for the 15 fountains in dry and wet periods.

**Figure 6 microorganisms-10-02458-f006:**
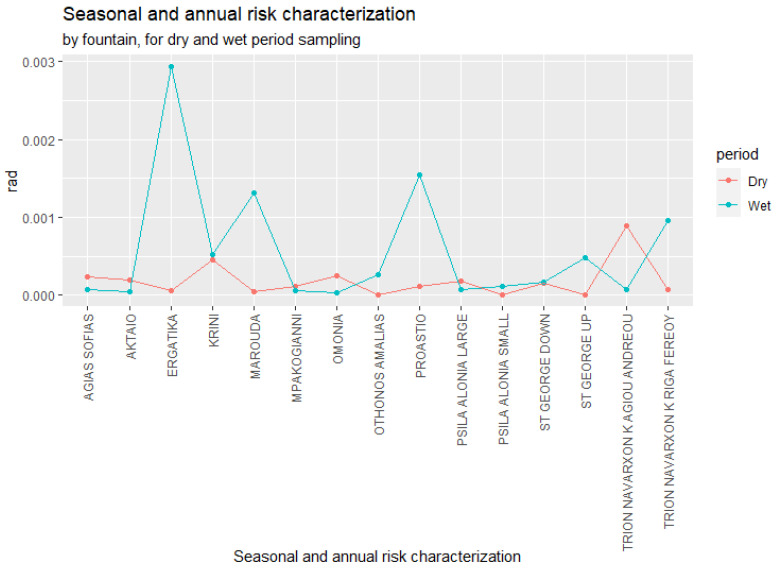
*Legionella* spp. seasonal risk characterization by fountain for wet and dry sampling periods.

**Table 2 microorganisms-10-02458-t002:** Model equations from Armstrong and Haas [28,31].

	QMRA Equations	Model Parameters
Bacterial water-to-air partitioning evaluation	Equation (1): C_air_ = C_water ×_ EF_fountains_	C_air_: Bacterial concentration in air (cfu/m^3^) C_water_: Bacterial concentration in water (cfu/Liter) EF_fountains_: Emission factor (liter/m^3^)
Inhalation exposure dose assessment	Equation (2): IED = C_air_ × ED × RR × IR	IED: Weekly inhaled exposure dose (cfu) ED: Weekly exposure duration (minutes) RR: Retention rate of aerosols in the lungs IR: Inhalation rate (m^3^/60 min)
Dose–response modeling for *Legionella* infections	Equation (3): R_w(d)_ = 1 − e^(−γd)^	R_w(d)_: Predicted risk given the weekly dose d: Weekly inhaled *Legionella* dose γ: Model parameter for *Legionella* infection risk = 0.06 (1/cfu)
Seasonal and annual risk characterization	Equation (4): R_a(d)_ = 1 − $\Pi_{w=1}^{n}$[1 − Rw(d)]	R_a(d)_: Seasonal risk n: Total number of exposure events

**Table 3 microorganisms-10-02458-t003:** The percentages of compliance of the fountains with the question points from the checklist (*n* = 15) for the dry and wet periods.

Scoring Items	% Dry Period	% Wet Period
1. The system operates according to the manufacturer’s instructions	100%	100%
2. The system is maintained in an acceptable condition—in the absence of waste, leaves, etc.	60%	67%
3. Absence of signs of development of corrosion, salts, microbiological growth	20%	73%
4. Absence of algae (optically)	33%	60%
5. Lack of leaks	100%	100%
6. Absence of obvious faults	93%	93%
7. Filters are maintained in good condition	-	-
8. Visual inspection of the system and diagram control do not indicate that there is stagnant water	93%	100%
9. There is no water reflux in the water supply system	100%	100%
10. The system is cleaned, drained and disinfected when out of service for more than a month	100%	100%
11. Chemicals are used to clean the salts	0%	0%
12. The water temperature is below 25 °C	20%	100%
13. The residual biocidal product concentration was found to be 0.4–0.7 mg/L if chlorine was used	0%	0%
14. The pH of the water is 7.0–8.0 if chlorine is used	93%	87%
15. At least one chemical test was carried out in the last year	0%	0%
16. At least one microbiological examination was performed in the last year by the relevant local authorities for public fountains	0%	0%
17. Sampling for microbiological testing by the laboratory	100%	100%

**Table 4 microorganisms-10-02458-t004:** *Legionella* spp. concentrations (cfu/L) in water (C_water_) and air (C_air_) for fountains during the dry and wet periods. The average Legionella concentrations were based on the Legionella counts. The average concentrations in the air (cfu/m^3^) were based on the water–air bacterial coefficients. The Legionella inhaled exposure doses (IEDs, cfu) from different fountains for each period are shown. Ra(d) was calculated based on the equations in Table 2.

Fountain	C_water_ Dry (cfu/L)	C_water_ Wet (cfu/L)	C_air_ Dry (cfu/m^3^)	C_air_ Wet (cfu/m^3^)	IED Dry	IED Wet	Ra(d) Dry	Ra(d) Wet
1	2550	1250	2.20 × 10^−5^	1.10 × 10^−5^	3.8 × 10^−3^	1.86 × 10^−3^	2.3 × 10^−4^	1.10 × 10^−4^
2	2150	0	1.80 × 10^−5^	0.00 × 10^0^	3.2 × 10^−3^	0.00 × 10^0^	1.9 × 10^−4^	0.00 × 10^0^
3	1650	800	1.40 × 10^−5^	6.90 × 10^−6^	2.5 × 10^−3^	1.19 × 10^−3^	1.5 × 10^−4^	7.20 × 10^−5^
4	9900	0	8.50 × 10^−5^	0.00 × 10^0^	1.48 × 10^−2^	0.00 × 10^0^	8.8 × 10^−4^	0.00 × 10^0^
5	2000	0	1.70 × 10^−5^	0.00 × 10^0^	3 × 10^−3^	0.00 × 10^0^	1.8 × 10^−4^	0.00 × 10^0^
6	2850	550	2.50 × 10^−5^	4.70 × 10^−6^	4.2 × 10^−3^	8.2 × 10^−4^	2.5 × 10^−4^	4.90 × 10^−5^
7	1250	600	1.10 × 10^−5^	5.20 × 10^−6^	1.9 × 10^−3^	8.9 × 10^−4^	1.1 × 10^−4^	5.40 × 10^−5^
8	5100	800	4.40 × 10^−5^	6.90 × 10^−6^	7.6 × 10^−3^	1.19 × 10^−3^	4.6 × 10^−4^	7.20 × 10^−5^
9	17,350	550	1.50 × 10^−4^	4.70 × 10^−6^	2.59 × 10^−2^	8.2 × 10^−4^	1.55 × 10^−3^	4.90 × 10^−5^
10	5300	1900	4.60 × 10^−5^	1.60 × 10^−5^	7.9 × 10^−3^	2.83 × 10^−3^	4.7 × 10^−4^	1.70 × 10^−4^
11	10,750	850	9.20 × 10^−5^	7.30 × 10^−6^	1.6 × 10^−2^	1.27 × 10^−3^	9.6 × 10^−4^	7.60 × 10^−5^
12	1300	750	1.10 × 10^−5^	6.40 × 10^−6^	1.9 × 10^−3^	1.12 × 10^−3^	1.2 × 10^−4^	6.70 × 10^−5^
13	2950	400	2.50 × 10^−5^	3.40 × 10^−6^	4.4 × 10^−3^	6 × 10^−4^	2.6 × 10^−4^	3.60 × 10^−5^
14	14,750	650	1.30 × 10^−4^	5.60 × 10^−6^	2.2 × 10^−2^	9.7 × 10^−4^	1.32 × 10^−3^	5.80 × 10^−5^
15	32,850	5900	2.80 × 10^−4^	5.10 × 10^−5^	4.89 × 10^−2^	8.79 × 10^−3^	2.93 × 10^−3^	5.30 × 10^−4^

## Data Availability

Data supporting the reported results are available upon request.

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
