# Peer review of "Surveillance of Legionella spp. in Open Fountains: Does It Pose a Risk?"

_microorganisms, 2022, doi:10.3390/microorganisms10122458_

Round 1

Reviewer 1 Report

A very valuable article. The issues of Legionella concentration and the risk of infection are important for society as a whole. Methodology presented in detail. The results are clear and well-discussed. Correct discussion. Conclusions result from the research and calculations carried out.

Author Response

Thanks for your kind and positive review.

Reviewer 2 Report

Dear Authors,

the manuscript reveals an interesting and original study, including QMRA based on the research results. Study design is well described and clear. 

Some clarifications and/ or figures to describe sampling in the fountains can be added to understand better both design of fountains and sampling itself!

Authors could modify and/ or clarify this part!

Regarding the results- it is not clear enough in the Table 3 in the list of scoring items- how absence or presence of algae were detected?/ defined?

Due to the high prevalence of Legionella in fountains Authors, could highlight also what kind of actions have to be taken by the health inspectorate or relevant agencies or legislative initiatives taken by national / EU authorities?

Overall Conclusions are clear, however most of the last paragraph of the Conclusion part could be moved to the Discussion thus reducing volume and improving the focus of the Conclusions.

In general, I think that the manuscript after minor revision and minor English check could be published as an interesting study for the research community!

Sincerely,

Reviewer

Author Response

Dear Reviewer

  1. English spell check applied as indicated by the reviewer
  2. Sampling procedure clarified according to the recommendation (L144-145)
  3. Regarding the results concerning theabsence or presence of algae, clarifications were added

  4. Actions to be taken by national/EU Authorities were added (L452-462)
  5. Last paragraph of conclusions transferred in the discussion part as indicated by the reviewer 

Kind regards

Ioanna P. Chatziprodromidou

Reviewer 3 Report

The study was interesting and sufficiently original. At the information derived from the check list something such as a possible closed water cycle of the fountains and observations about the presence/absebnce of biofilms should be added.

Author Response

Dear Reviewer,

thanks for your kind review.

  1. We proceeded to spell check from a native speaker
  2. Info concerning absence/presence of biofilms added as indicated by the reviewer
